# Chemomechanical modification of quantum emission in monolayer WSe$_2$

M. Iqbal Bakti Utama [1,10], Hongfei Zeng [2,10], Tumpa Sadhukhan [3,9,10], Anushka Dasgupta [1,10], S. Carin Gavin[2], Riddhi Ananth[3], Dmitry Lebedev[1], Wei Wang [4], Jia-Shiang Chen [4,5], Kenji Watanabe [6], Takashi Taniguchi [7], Tobin J. Marks [1,3], Xuedan Ma [4,5], Emily A. Weiss [3], George C. Schatz [3] ✉, Nathaniel P. Stern [2] ✉ & Mark C. Hersam [1,3,8] ✉

Two-dimensional (2D) materials have attracted attention for quantum information science due to their ability to host single-photon emitters (SPEs). Although the properties of atomically thin materials are highly sensitive to surface modification, chemical functionalization remains unexplored in the design and control of 2D material SPEs. Here, we report a chemomechanical approach to modify SPEs in monolayer WSe$_2$ through the synergistic combination of localized mechanical strain and noncovalent surface functionalization with aryl diazonium chemistry. Following the deposition of an aryl oligomer adlayer, the spectrally complex defect-related emission of strained monolayer WSe$_2$ is simplified into spectrally isolated SPEs with high single-photon purity. Density functional theory calculations reveal energetic alignment between WSe$_2$ defect states and adsorbed aryl oligomer energy levels, thus providing insight into the observed chemomechanically modified quantum emission. By revealing conditions under which chemical functionalization tunes SPEs, this work broadens the parameter space for controlling quantum emission in 2D materials.

Two-dimensional (2D) van der Waals materials, such as hexagonal boron nitride (hBN) and transition metal dichalcogenides (TMDs), have been widely explored as hosts for single-photon emitters (SPEs)[1–8]. In particular, the combination of SPEs and valley pseudospin physics[9] in monolayer (1L) WSe$_2$ makes this 2D material especially attractive for the transduction of quantum information from spin-related degrees of freedom into single photons. Consequently, methods for controlling and modifying SPEs in 1L WSe$_2$ are critical to applications in quantum information science, such as quantum communication[10]. Thus far, the vast majority of TMD SPE research has explored only a single mechanism for manipulating SPEs either by using localized strain for exciton funneling[11–13] (such as nanopillar[12,13] or nanoindentation[14] arrays) or by performing defect engineering for exciton trapping[15,16]. Even when these two mechanisms have been used in tandem[17], limited tunability has thus far been achieved in the resulting quantum emission characteristics.

One common observation in the SPE properties of WSe$_2$ is a complicated low-temperature spectrum with many emission lines[1,3,4,11,18]

[1]Department of Materials Science and Engineering and the Materials Research Center, Northwestern University, Evanston, IL 60208, USA. [2]Department of Physics and Astronomy, Northwestern University, Evanston, IL 60208, USA. [3]Department of Chemistry and the Materials Research Center, Northwestern University, Evanston, IL 60208, USA. [4]Center for Nanoscale Materials, Argonne National Laboratory, Lemont, IL 60439, USA. [5]Northwestern-Argonne Institute of Science and Engineering, Northwestern University, Evanston, IL 60208, USA. [6]Research Center for Functional Materials, National Institute for Materials Science, 1-1 Namiki, Tsukuba 305-0044, Japan. [7]International Center for Materials Nanoarchitectonics, National Institute for Materials Science, 1-1 Namiki, Tsukuba 305-0044, Japan. [8]Department of Electrical and Computer Engineering, Northwestern University, Evanston, IL 60208, USA. [9]Present address: Department of Chemistry, SRM Institute of Science and Technology, Kattankulathur, Tamil Nadu 603203, India. [10]These authors contributed equally: M. Iqbal Bakti Utama, Hongfei Zeng, Tumpa Sadhukhan, Anushka Dasgupta. ✉e-mail: g-schatz@northwestern.edu; n-stern@northwestern.edu; m-hersam@northwestern.edu

due to the complex defect landscape within samples that trap excitons[18,19]. Although spectrally- and spatially isolated emitters in WSe₂ exhibiting non-classical photon behavior have been reported even from crowded emission spectra, a high density of emission lines around an emitter of interest can create a challenge to completely filter out signals from neighboring emitters and the broad defect background, impacting the purity of single photons extracted from such a spectrum. A high density of these emission lines is hence undesirable for quantum transduction experiments. This issue provides the impetus for investigating alternative strategies for controlling SPE properties in WSe₂. Since chemical functionalization has been shown to be an effective strategy for tuning the electronic and optical properties of 2D semiconductors[20], this approach is also likely to be useful for tuning quantum emission, especially because interfacial modulation is known to strongly influence excitonic properties[21]. Despite this promise, surface and interface engineering via chemical functionalization has not yet been successfully employed to tune SPEs in TMDs.

Here, we report a chemomechanical modification approach for 1L WSe₂ that produces spectrally isolated SPEs via a synergistic combination of localized mechanical strain and chemical functionalization using aryl diazonium chemistry. In particular, surface modification of strained 1L WSe₂ with 4-nitrobenzenediazonium (4-NBD) tetrafluoroborate quenches most strain-induced defect emission, resulting in sharp SPEs with high single-photon purity. Rather than covalently reacting with WSe₂, the 4-NBD treatment conditions result in a physisorbed nitrophenyl oligomer layer on the WSe₂ surface as confirmed by X-ray photoelectron spectroscopy, atomic force microscopy, and photoluminescence imaging. First-principles calculations show that shallow mid-gap states from the physisorbed nitrophenyl oligomer layer are energetically resonant with WSe₂ defect levels, thereby suppressing most emission pathways and simplifying the final SPE spectrum. Overall, these results establish chemical functionalization as an effective pathway for modifying SPE in strained 1L WSe₂.

## Results

### Diazonium functionalization and photoluminescence quenching

Figure 1a schematically depicts the spontaneous chemical functionalization that occurs upon immersion of 1L WSe₂ into an aqueous solution of 4-NBD tetrafluoroborate. The electrophilic nature of 4-NBD cations is believed to withdraw electrons from WSe₂, releasing N₂ and generating diazonium radicals in addition to causing hole doping in WSe₂[22]. Although aryl diazonium radicals are often assumed to subsequently form covalent bonds with the 2D material surface[22-29], the highly reactive diazonium radicals can also react with one another, forming nitrophenyl (NPh) oligomer chains of varying lengths[30], including the 2-ring and 3-ring structures illustrated in Fig. 1a (also see Supplementary Fig. 1b–d for a general functionalization scheme using 4-NBD). As will be discussed in more detail below, our reaction conditions favor oligomerization, resulting in a physisorbed NPh adlayer on the 1L WSe₂ surface. In particular, following immersion for 1.5 h in a 5 mM aqueous solution of 4-NBD (Supplementary Fig. 2), 1L WSe₂ is fully coated with a ~4–5 nm thick NPh oligomer adlayer that quenches its room-temperature photoluminescence (PL). To illustrate this point, a 1L WSe₂ sample was prepared that is partially covered with an hBN flake (Fig. 1b). Figure 1c shows the PL spectrum collected from the 1L WSe₂ region without the hBN cover before and after the 4-NBD treatment, where the integrated PL drops to ~15% of its original level following chemical modification. Meanwhile, PL of the hBN-covered WSe₂ region is not quenched (Supplementary Fig. 3c) since the hBN cover prevents direct contact between the NPh oligomers and the WSe₂ surface. This PL quenching effect is also evident by comparing the spatial map of the peak PL intensity in Fig. 1d, g. In addition, although the PL spectrum of the hBN-covered region shows no significant changes (Supplementary Fig. 3c), the PL of the uncovered WSe₂ following 4-NBD treatment shows a PL peak redshift by ~11 meV (Fig. 1e, h) and linewidth broadening (Fig. 1f, i).

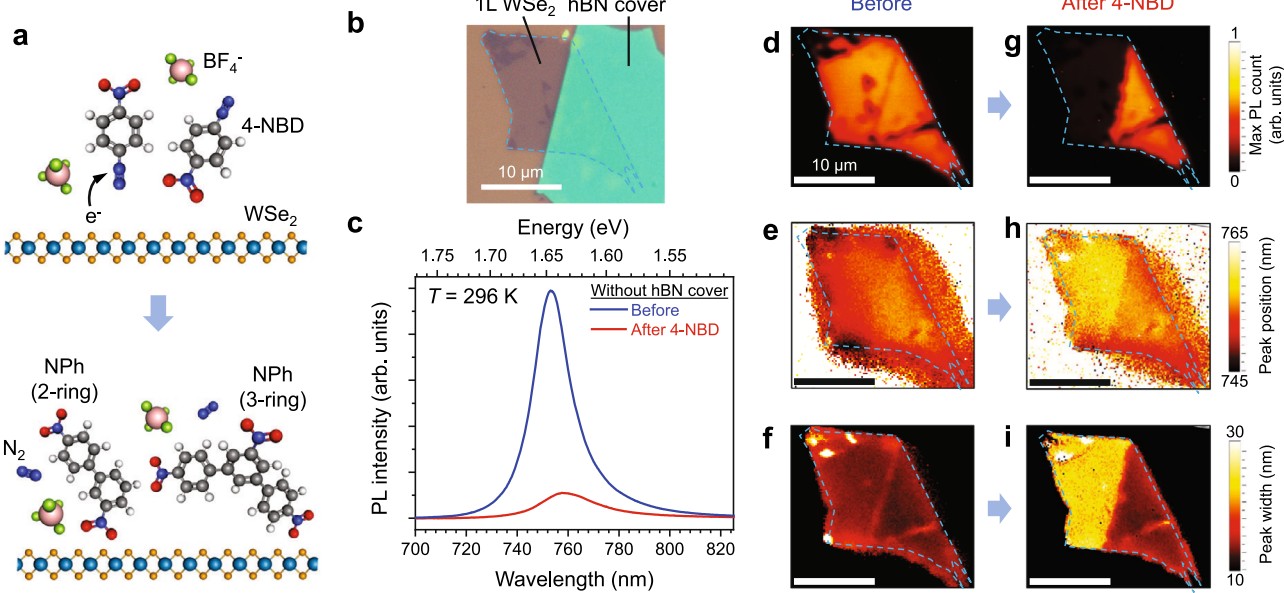

**Fig. 1 | Room-temperature photoluminescence (PL) quenching following 4-nitrobenzenediazonium (4-NBD) treatment. a** Illustration of the chemical functionalization scheme. **b** Optical image of a 1L WSe₂ that is partially covered with a thin hBN flake. **c** Room-temperature (T = 296 K) PL spectra of 1L WSe₂ before and after 4-NBD treatment from a location without the hBN cover, showing PL intensity quenching, redshifting, and broadening. The PL spectrum from hBN-covered WSe₂ does not show significant changes before and after 4-NBD treatment (Supplementary Fig. 3) and has been used to normalize the PL intensity. **d–i** PL map of the sample before (**d–f**) and after the 4-NBD treatment (**g–i**). The color in each map represents: **d, g** the max PL count, **e, h** peak position, **f, i** and peak width. While the hBN-covered region shows negligible change with 4-NBD treatment, the uncovered region exhibits PL quenching, redshifting, and broadening. Scale bars in (**d–i**) correspond to 10 μm.

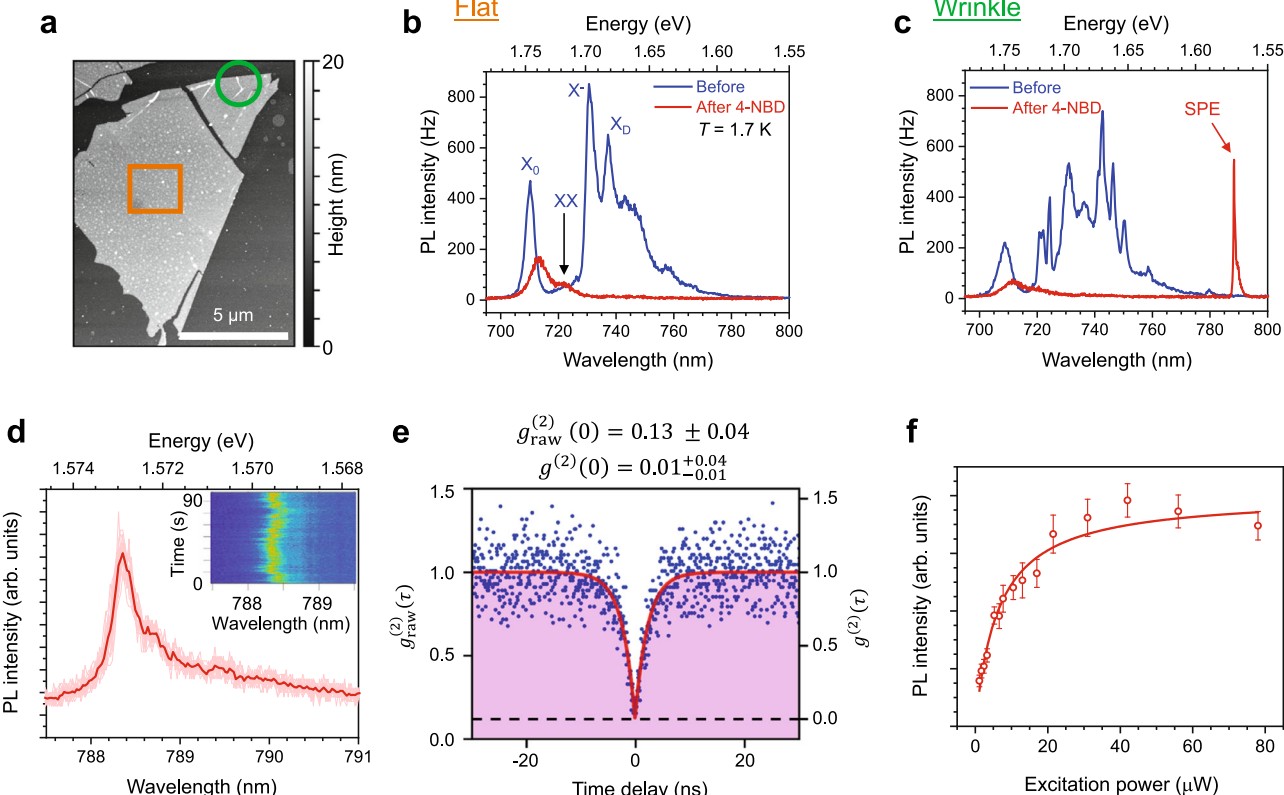

**Fig. 2 | Partial quenching of defect emitters in locally strained 1L WSe₂. a** Height image from atomic force microscopy (AFM) of a 1L WSe₂ flake. The region marked with the orange square is relatively flat, whereas the region inside the green circle contains wrinkles with localized strain. **b, c** Low-temperature ($T = 1.7$ K) photo-luminescence (PL) spectra of 1L WSe₂ at (**b**) the flat region and (**c**) the wrinkle, before (blue) and after (red) 4-nitrobenzenediazonium (4-NBD) treatment. The spectra were measured with ~60 μW excitation power. **d** High-resolution spectrum of the single-photon emitter (SPE) from (**c**). The thick red trace is the time average of the traces shown in pink. Inset: SPE spectral diffusion plot. **e** Second-order correlation function of the SPE. The left vertical axis denotes the as-measured $g_{\mathrm{raw}}^{(2)}(\tau)$ values, whereas the right vertical axis denotes the background-corrected $g^{(2)}(\tau)$ values. Fitting of the data (red curve) reveals a $g_{\mathrm{raw}}^{(2)}(0)$ of $0.13 \pm 0.04$, while background correction results in a $g^{(2)}(0)$ of $0.01_{-0.01}^{+0.04}$ (the details of the background correction are available in "Methods"). The black dashed line marks where $g^{(2)}(\tau) = 0$ after background correction. **f** SPE intensity as a function of excitation power. The error bars represent the standard deviation from the time averaging and the red solid line is a fit to the data.

## Low-temperature optical spectroscopy

Figure 2 explores the low-temperature optical properties of a 1L WSe₂ flake on a flat Si/SiO₂ substrate. Atomic force microscopy (AFM) shows that the sample is generally flat, although the flake has wrinkles near the edges as indicated by the green circle in Fig. 2a. Before 4-NBD functionalization, the PL spectrum collected from the flat area of the 1L WSe₂ sample possesses rich excitonic features (Fig. 2b, blue curve), which are assigned following previous literature[31,32] as the neutral A exciton ($X_0$), biexciton (XX), negatively-charged trion ($X^-$), and dark exciton ($X_D$). Subsequent 4-NBD treatment quenches most of these spectral features, leaving $X_0$ as the remaining dominant emission feature (Fig. 2b, red curve). Unlike diazonium functionalization of carbon nanotubes[33–36], no SPEs or new spectral features are observed on the flat region of the chemically modified 1L WSe₂. The absence of lower-energy positively-charged trion ($X^+$) emission features after chemical modification, as has been observed for 1L WSe₂ that was hole-doped by electrostatic gating[32,37], suggests that the quenching effect of the 4-NBD treatment cannot be solely attributed to hole doping. Likewise, in the alternative scenario that 4-NBD treatment changes the doping level only close to charge neutrality, the PL quenching also cannot be attributed to functionalization-induced doping compensation because the neutral exciton usually appears with even higher PL intensity (instead of being quenched) when the doping level is brought closer to charge neutrality[37].

Unlike the flat regions, the wrinkled location on the 1L WSe₂ flake preceding chemical modification shows spectrally dense emission features between 720 nm and 780 nm that are brighter than the original neutral exciton (Fig. 2c, blue). This low-temperature PL enhancement at wrinkles is commonly observed and has been attributed to a strain-assisted hybridization of dark exciton and defect states that increases the radiative recombination yield[38]. The 4-NBD treatment effectively quenches these defect-related emission features (Fig. 2c, red), resulting in significantly fewer emission features that are spectrally isolated (e.g., the sharp peak near 788 nm). As we shall discuss later, it is likely that the emission features remaining after 4-NBD treatment also originate from pre-existing defects within the WSe₂ monolayer. A high-resolution spectrum of this sharp feature (Fig. 2d) reveals a zero-phonon linewidth of ~0.5 meV. Observing the spectrum over time shows that this emitter is relatively stable without significant spectral diffusion (Fig. 2d, inset). The time series of the peak position can be constructed into a histogram of the spectral jitter and fitted with a Gaussian distribution, resulting in jitter FWHM of 200-400 μeV (Supplementary Fig. 4).

The second-order correlation function, $g^{(2)}(\tau)$, of the emitter state shows clear antibunching behavior (Fig. 2e). Fitting the measurement with a two-level model using the equation $g^{(2)}(\tau) = 1 - ae^{-|\tau|/\tau_1}$, where $a$ is a constant, yields a raw $g_{\mathrm{raw}}^{(2)}(\tau) = 0.13 \pm 0.04$. Background correction results in $g^{(2)}(0) = 0.01_{-0.01}^{+0.04}$, which is significantly lower than the widely accepted 0.5 threshold for SPE, thus indicating high purity of the produced single photons (the background-corrected $g^{(2)}(\tau)$ is shown in the right y-axis). The fitting also allows extraction of the emission lifetime, $\tau_1 = (2.34 \pm 0.13)$ ns, which is comparable to the typically

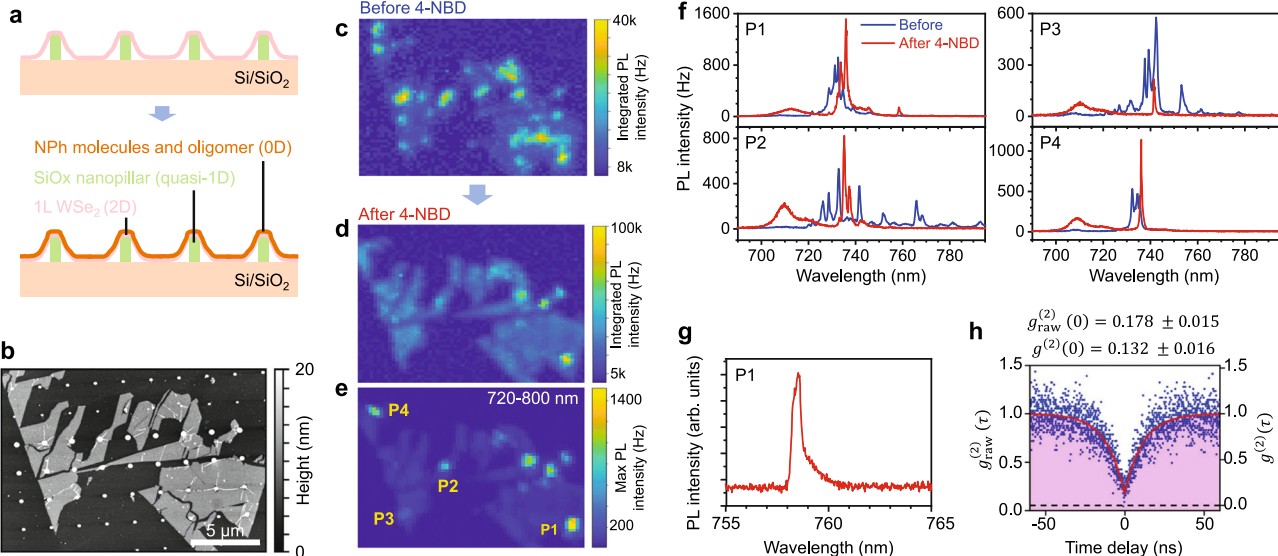

**Fig. 3 | Chemomechanically modified emitters on a nanopillar array.**
**a** Schematic of the mixed-dimensional heterostructures consisting of consisting of nitrophenyl (NPh) oligomers (0D), nanopillars (quasi-1D), and 1L WSe$_2$ (2D) that also host the quantum emitter (0D). **b** Atomic force microscopy (AFM) image of a 1L WSe$_2$ flake on a nanopillar array after 4-nitrobenzenediazonium (4-NBD) treatment. **c** Low-temperature ($T = 1.7$ K) photoluminescence (PL) map of integrated intensity of the as-transferred flake before the 4-NBD treatment. **d**, **e** PL maps after the 4-NBD treatment, showing (**d**) integrated intensity over the measured spectral range and (**e**) maximum count for wavelengths between 720 nm and 800 nm only. **f** PL spectra at selected nanopillar sites marked in (**e**) before (blue) and after (red) 4-NBD treatment. The 4-NBD treatment simplifies the emission spectra by quenching most of the dense defect-related emission. **g** Single-photon emitter (SPE) after the 4-NBD treatment at location P1. **h** Second-order correlation function measurement on the SPE in (**g**). The $g^{(2)}_{raw}(\tau)$ values are on the left vertical axis, whereas the background-corrected $g^{(2)}(\tau)$ values are on the right vertical axis.

observed value for 1L WSe$_2$ SPE in the literature[4,5] and the lifetime measured using time-resolved fluorescence (Supplementary Fig. 5 and Supplementary Table 1). This emitter also shows a saturating power dependence that is typical for 1 L WSe$_2$ SPE (Fig. 2f). Fitting the excitation power dependence of the PL intensity with the equation $I = I_{sat}P/(P + P_{sat})$ yields a saturation power ($P_{sat}$) of $(6.3 \pm 0.7)$ μW. Because the emitter only appears on the wrinkled regions of the chemically modified 1L WSe$_2$ flake and has properties consistent with strained 1L WSe$_2$ SPE, this SPE likely originates from the WSe$_2$ itself rather than from the NPh film alone or from interlayer excitonic species between the molecular film and the WSe$_2$ flake. It should also be noted that resonance enhancement is observed as the excitation energy passes through the WSe$_2$ A exciton resonance (Supplementary Fig. 6), further confirming that this SPE originates from the 1L WSe$_2$ flake.

**Spatially deterministic chemomechanically modified SPE**

In an effort to translate the chemomechanically modified SPE from random wrinkled regions to spatially deterministic locations, 1L WSe$_2$ was transferred onto an array of prefabricated SiO$_x$ nanopillars (Fig. 3a, also see Supplementary Figs. 7–8), resulting in mixed-dimensional heterostructures[39] consisting of NPh oligomers (0D), nanopillars (quasi-1D), and 1L WSe$_2$ (2D) that also host the quantum emitters (0D). By comparing the AFM image (Fig. 3b) and low-temperature PL map of the sample (Fig. 3c–e), the location of each nanopillar is determined, thereby enabling comparison of the emission before and after 4-NBD treatment. Similar to 1L WSe$_2$ on a flat substrate, chemical functionalization quenches the majority of dense defect-related emission lines between 730 nm and 760 nm. The 4-NBD treatment also quenches the series of sharp defect-related emission features that are initially brightened by the strain induced by the nanopillars.

Comparison of the PL spectra at the same nanopillar location (e.g., P1-P4 in Fig. 3f) reveals the simplification of the emission spectra following the 4-NBD treatment, where significantly fewer, energetically isolated emitter states remain (also see Supplementary Figs. 9–10 for spectra from other locations on the sample). One example is the

sharp emission peak at ~759 nm for nanopillar position P1 (Fig. 3g). From $g^{(2)}(\tau)$ measurements on this emission feature, clear antibunching is observed with a raw $g^{(2)}_{raw}(0)$ value of $0.178 \pm 0.015$, background-corrected $g^{(2)}(0)$ value of $0.132 \pm 0.016$, and $\tau_1 = (11.5 \pm 0.5)$ ns, thereby confirming high-purity SPE following 4-NBD functionalization. Supplementary Figs. 11 and 12 provide the statistics for the emitter properties observed with our chemomechanical modification scheme, including the yield, peak wavelength distribution, peak PL intensity, and narrowest linewidth. Meanwhile, Supplementary Fig. 13 provides a measurement of a monolayer WSe$_2$ flake before and after 4-NBD treatment at low temperature with identical excitation conditions.

**Noncovalent functionalization following 4-NBD treatment**

To investigate the nature of the bonding between the molecular adlayer and WSe$_2$ following the 4-NBD treatment, surface-sensitive characterization was performed using X-ray photoelectron spectroscopy (XPS). Figure 4a, b shows XPS spectra on WSe$_2$ before and after the 4-NBD treatment. For both the W and Se core levels, a chemical shift of ~0.5 eV to a higher binding energy is observed after chemical functionalization. A similar shift following 4-NBD treatment on WSe$_2$ has previously been attributed to hole doping, which causes the Fermi level to be displaced closer to the valence band maximum and the core levels[22]. However, the XPS spectra do not show evidence of new chemical bond formation since no new spectral features nor lineshape changes are observed (Supplementary Figs. 14 and 15) that can be attributed to the formation of Se−C or W−C bonds. Likewise, no lineshape changes are apparent even when the sample is tilted by as much as 40° to improve the surface sensitivity of the XPS measurement (Supplementary Fig. 16). These XPS results suggest that the 4-NBD treatment results in a physisorbed NPh oligomer adlayer without chemical bond formation to the WSe$_2$ surface.

Corroborating the physisorbed nature of the molecular adlayer, the NPh oligomer film is easily removed without damaging the underlying WSe$_2$ using contact-mode AFM. In particular, AFM

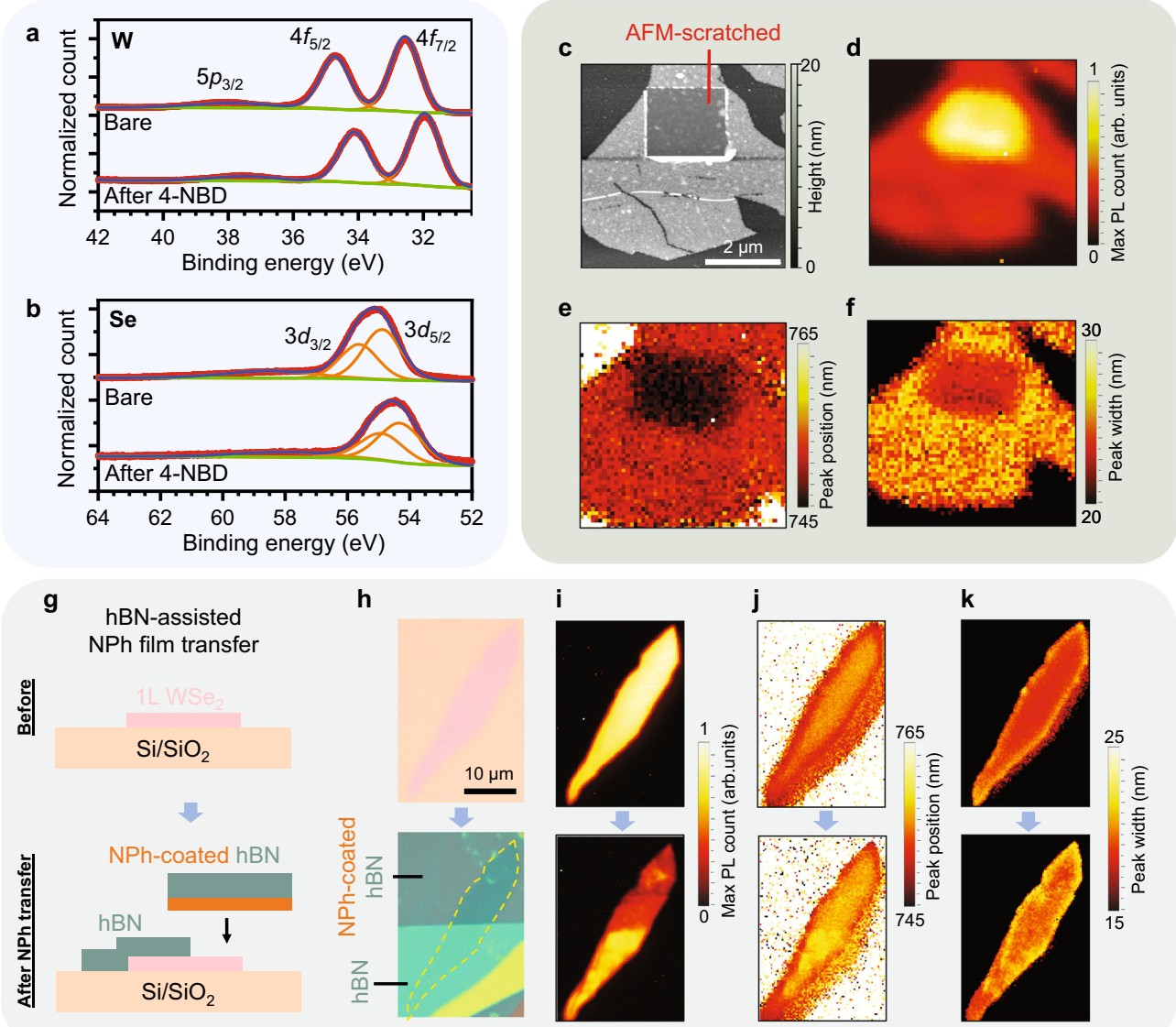

**Fig. 4 | Noncovalent NPh functionalization of WSe₂. a, b** X-ray photoelectron spectroscopy (XPS) spectra of WSe₂ after 4-nitrobenzenediazonium (4-NBD) treatment for (**a**) W and **b** Se core levels. While the chemical shift to higher binding energies can be explained by hole doping, no lineshape changes are observed that would be indicative of bond formation between WSe₂ and nitrophenyl (NPh). Red: experimental data points, orange: fitted peaks, blue: sum of fitted peaks, green: baseline. **c–f** Reversible quenching of the WSe₂ PL with AFM scratching. **c** Atomic force microscopy (AFM) image of 1L WSe₂ after removal of a $2 \times 2 \, \mu m^2$ region of the NPh film using contact-mode AFM. The PL map of the sample is shown for **d** the maximum count, **e** the peak position, and **f** the peak width. **g–k** PL quenching on 1L WSe₂ functionalized with a NPh film via hBN-assisted transfer. **g** Schematic of the NPh film transfer process. **h** Optical micrograph of as-exfoliated WSe₂ before and after NPh film transfer. The photoluminescence (PL) map before and after the transfer is shown for **i** peak intensity, **j** peak position, and **k** peak width.

scratching with a normal force setpoint between 50 nN and 200 nN results in nearly complete removal of the molecular adlayer (Fig. 4c), allowing the NPh oligomer film thickness of ~4.4 nm to be directly measured (Supplementary Fig. 17). AFM scratching also results in reversal of the effects of the 4-NBD treatment on the room-temperature 1L WSe₂ PL spectrum as the PL intensity in the scratched region returns to its higher value relative to the surrounding area that remains coated with NPh (Fig. 4d). Furthermore, the PL peak is blue-shifted (Fig. 4e) and the peak width is decreased (Fig. 4f) following AFM scratching, as expected for pristine 1L WSe₂. The change of the PL lineshape following 4-NBD treatment is also reversed upon removal of the NPh film (see Supplementary Figs. 17e, f and Fig. 18c for the PL spectra).

To further demonstrate that the PL spectral changes following the 4-NBD treatment are not resulting from covalent surface modification, a NPh film was transferred onto the surface of WSe₂ using a carrier hBN flake (Fig. 4g, h; also see Supplementary Figs. 19–20). This NPh transfer

method circumvents the need for WSe₂ immersion into the 4-NBD tetrafluoroborate solution, thereby avoiding WSe₂ exposure to diazonium radicals and ensuring a noncovalent interaction between the NPh film and the WSe₂ surface (Supplementary Fig. 1f, g). The PL map obtained after NPh film transfer shows quenching of the neutral exciton emission on the upper half of the 1L WSe₂ that is in direct contact with the NPh film (Fig. 4i). However, unlike direct exposure of WSe₂ to the 4-NBD solution, the WSe₂ PL spectrum for the transferred NPh film does not show an appreciable redshift nor broadening of the linewidth (Fig. 4k; also see Supplementary Fig. 21). The observed PL quenching cannot be attributed to interference effects or local field changes due to the variation in hBN thickness, as shown in our calculation of the light emission outcoupling from the sample (Supplementary Fig. 22). Instead, the weaker quenching in the sample with transferred NPh can be explained by the interface of WSe₂ and transferred NPh film that is less conformal than the typical interface of NPh/WSe₂ from direct

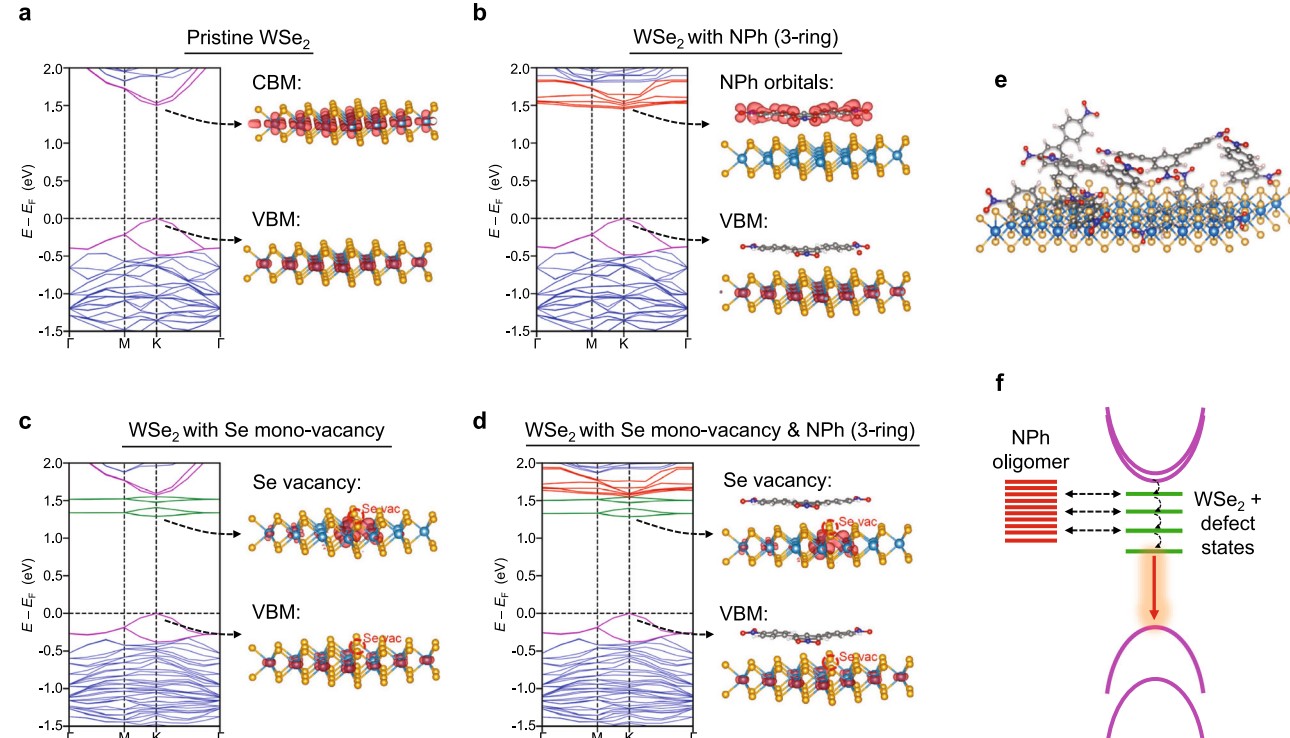

**Fig. 5 | Density functional theory (DFT) calculations for 1L WSe₂ and nitrophenyl (NPh) oligomers.** The band structure calculations presented here assume a 1% biaxial tensile strain for the 1L WSe₂ substrate. The Fermi energy ($E_F$) is set to the energy of valence band maximum (VBM). **a** Pristine, defect-free 1L WSe₂. Here, blue: electronic bands from WSe₂, purple: spin–orbit split bands of WSe₂ that are closest to the VBM and conduction band maximum (CBM) band edges. **b** 1L WSe₂ with a physisorbed NPh 3-ring oligomer. The real-space distribution of the density of states shows that the valence band maximum retains the WSe₂ character while the lowest energy of the weakly dispersive mid-gap states near the conduction band minimum (CBM) are localized at the NPh oligomer, suggesting the formation of a type-II heterojunction. Although the change in the WSe₂ conduction bands is minimal, there is a finite coupling of the bands to the higher lying NPh orbitals, and thus the conduction bands near CBM are also colored in red. **c** 1L WSe₂ with a single

mono-vacancy of Se, as a prototypical example of defects that can emit light with localized strain. The new mid-gap states near the CBM are localized at the defect site and are colored in green. **d**, Calculation for the combination of a Se mono-vacancy and NPh 3-ring oligomer. **e** Illustration of NPh oligomers physisorbed on the surface of 1L WSe₂. In a typical sample, the 4-NBD treatment results in a polydisperse mixture of NPh oligomer species. **f** Illustration of the quenching mechanism that results in a simplified SPE spectrum following 4-NBD treatment. The represented colors of the band in this illustration follow the color coding used in the calculation in (**a**–**d**). Black dashed arrow indicates NPh orbitals and WSe₂ defect states that are in resonance and can effectively quench the emission from these defects. The strain- and defect-trapped exciton thus only recombine radiatively from the available defect states with lower energy that are not quenched by the NPh states (red arrow).

solution phase treatment with 4-NBD (Supplementary Fig. 23). Overall, these results suggest that the WSe₂ PL quenching following the 4-NBD treatment cannot be primarily attributed to hole doping or covalent bonding, necessitating the identification of an alternative mechanism using first-principles calculations.

## First-principles calculations

Density functional theory calculations were performed under the assumptions that the NPh film is physisorbed on the WSe₂ surface and that the NPh oligomers are randomly distributed in terms of ring configuration and chain length. In addition, the 1L WSe₂ substrate was subjected to a biaxial strain of 1% to simulate the experimental conditions when SPE was observed. Figure 5a shows the calculated band structure of pristine, defect-free 1L WSe₂ in addition to the charge density distributions in real-space for the valence band maximum (VBM) and the conduction band minimum (CBM). Multiple NPh configurations from 1 to 3 rings and possible open-shell and closed-shell systems were considered (Supplementary Fig. 24). These results show that the most energetically stable configuration is with the oligomer rings positioned flat to the 1L WSe₂ surface at an equilibrium distance of ~3.5 Å between the NPh oligomers and the WSe₂ surface (Supplementary Fig. 25).

With the 3-ring NPh oligomer as an example (Fig. 5b), DFT calculations indicate that physisorbed NPh oligomers on WSe₂ result in

nearly flat mid-gap states near the CBM of 1L WSe₂. Real-space charge density distributions show that these shallow mid-gap states primarily reside within the NPh oligomers. Comparison to the energy levels of isolated NPh (Supplementary Fig. 26) suggests that these mid-gap states originate from the lowest unoccupied molecular orbital (LUMO) and higher unoccupied orbitals of NPh. Meanwhile, the valence band maximum still resembles that of pristine 1L WSe₂, which indicates that a type-II heterojunction is formed between NPh and 1L WSe₂. DFT calculations for oligomers with different ring numbers and configurations show a diversity in the energies of the mid-gap states, which generally also form type-II heterojunctions with WSe₂ (Supplementary Figs. 26 and 27 and Supplementary Table 2). The formation of type-II heterojunctions explains the general PL quenching for 1L WSe₂ after 4-NBD treatment, including in the case without applied strain. Although the physisorbed NPh oligomers form a multilayered film, the NPh oligomers that are in direct contact with the WSe₂ surface likely play a dominant role in determining the heterojunction behavior. The linewidth broadening of the neutral exciton emission after functionalization with NPh is also consistent with the formation of a type-II heterojunction, where the lifetime of excited states is shortened by additional decay channels that lead to exciton dissociation[40].

DFT calculations were also performed for 1L WSe₂ point defects, with the chalcogen vacancy chosen as a prototypical example since it is believed to be one of the defects responsible for SPE[17,38,41]. For 1L WSe₂

with a Se mono-vacancy (Fig. 5c), new mid-gap states emerge near the CBM. The charge distribution of this state is predominantly localized on the three W atoms surrounding the Se vacancy. The combination of a Se mono-vacancy and physisorbed NPh (Fig. 5d) shows characteristics that combine Fig. 5b and c in the CBM region, with a 0.02 eV shift in band gap compared to Fig. 5c. It should also be noted that 1% biaxial strain does not change the band structure significantly compared to the unstrained case (calculations without strain are also available in Supplementary Figs. 28–30), other than small changes in the band gap value (increasing it by ~0.08 eV from the unstrained value) and slightly more coupling between localized and delocalized states (also see Supplementary Fig. 31). Overall, the polydispersity of the NPh oligomer structure in the experimental molecular adlayer (Fig. 5e) translates into a diversity of mid-gap states that are in resonance with most of the WSe$_2$ defect states, resulting in quenching of the vast majority of strain-activated SPE (Fig. 5f) such that any remaining SPEs will be energetically isolated with high photon purity, as is observed experimentally.

To elaborate further, new emitters that emerge after the 4-NBD treatment appear to show properties similar to typical WSe$_2$ emitters and thus likely originate from pre-existing defect states within the WSe$_2$ monolayer itself. However, preceding the 4-NBD treatment, many of these defects may have been outcompeted by other defects in the same vicinity that dominate the emission signal. The insight from the first-principles calculations suggests a mechanism for quenching of emission due to charge or energy transfer to the molecular orbitals of nitrophenyl oligomers, particularly when the orbital energies are resonant with or energetically positioned lower than the defect states. In addition, at some sites, additional defect states can exist whose energies are lower than the molecular orbitals such that quenching of emission from these defect states is not effective. Prior to functionalization, these lower-energy defect states may not produce significant photoluminescence due to the presence of higher energy defects that provide dominant emission pathways, but the lower-energy defect states then become the preferred states for emissive recombination after 4-NBD functionalization quenches emission from the higher energy defects.

## Discussion

This study has demonstrated that chemomechanical modification significantly simplifies the low-temperature PL spectrum of strained 1L WSe$_2$. Detailed surface characterization shows that 4-NBD solution processing results in a physisorbed NPh oligomer adlayer, which generates mid-gap states near the CBM of 1L WSe$_2$ as determined by DFT. Since these NPh mid-gap states are in resonance with WSe$_2$ defect states, the remaining SPEs are energetically isolated with high photon purity. It is likely that the chemomechanical strategies introduced here can be applied to other low-dimensional semiconductors, thus allowing the preparation of high-purity SPEs in other spectral ranges. While the 4-NBD treatment shown here is effective for different strain environments ranging from random wrinkles to spatially deterministic nanopillar arrays, the exact wavelength of the resulting energetically isolated SPEs varies from sample to sample and even from nanopillar to nanopillar in a given sample, which suggests that further improvements can be gained from alternative functionalization chemistries[42] and/or better control over the defects and strain in 1L WSe$_2$. Regarding the former approach, one key future direction is to achieve control over the energetic alignment of the functional group orbitals and the WSe$_2$ defect levels. For functionalization with 4-NBD, this goal may be achieved by controlling the oligomer formation and configuration. For example, 4-NBD functionalization via electrochemical reduction has produced controlled nitrophenyl monolayers without spontaneous oligomerization[43]. The choice of the diazonium aryl group is another parameter that can control the energetic alignment of the molecular orbitals[44], while also presenting opportunities to exploit unique features such as steric hindrance (e.g., 3,5-bis-tert-butyl benzenediazonium[45]) that can dictate the molecular configuration.

The chemical functionalization approach is compatible with other schemes and strategies that have been previously described in the literature to achieve state-of-the art emitters in terms of brightness and linewidth. This compatibility is facilitated by the fact that the functionalization is performed on only one side of the WSe$_2$ monolayer surface. Specific to the 4-NBD treatment, since the linewidth and brightness of emitters before and after functionalization are comparable, it is likely that improvements to the emitter properties on the sample before functionalization would be maintained after the 4-NBD treatment. For example, reduction in the emitter linewidth can be achieved by using alternative substrates and straining structure with lower defect and interface states than that of SiO$_2$ used here. Previous work that has demonstrated narrow linewidths reaching <100 μeV in WSe$_2$ includes the use of Al$_2$O$_3$ (which can then be proximitized with metallic structures for plasmonic coupling)[46,47], InGaP[48] or hBN[17,49] substrates or encapsulation. Meanwhile, enhancement of the emitter brightness can be achieved by integrating the chemomechanically modified emitters with plasmonic structures[46,50] or by relying on enhancement effects with photonic structures[51–53]. Alternatively, further permutations of the chemomechanical strategy may also achieve improvements in the emitter properties by exploring other combinations of molecules and 2D materials. For example, selecting molecules with precisely aligned energy levels could improve the selectivity of the emitter wavelength toward a longer wavelength range or to a very narrow spectral range, which could be valuable in reducing the inhomogeneous broadening of these emitters. Likewise, improvement in emitter brightness may be achieved by selecting molecules that form type-I heterojunctions with 2D materials[54].

## Methods

### Sample preparation

The WSe$_2$ flakes were micromechanically exfoliated from bulk single crystals using the standard scotch tape method. After identifying monolayer flakes on PDMS stamps, 1L WSe$_2$ was transferred onto Si with 285-nm-thick SiO$_2$ (Fig. 1 and Supplementary Fig. 3), PMMA-coated Si/SiO$_2$ (Supplementary Fig. 18), or prefabricated nanopillar arrays (Fig. 3) using a transfer stage inside a N$_2$ glove box (<0.1 ppm H$_2$O and O$_2$) following the conventional viscoelastic transfer method at room temperature[12,13,55].

### Chemical functionalization

The 4-nitrobenzenediazonium (NBD) tetrafluoroborate powder (97%, Sigma-Aldrich) was purified using recrystallization to remove impurities. The WSe$_2$ samples were then immersed into a 5 mM 4-NBD tetrafluoroborate aqueous solution for 30–120 min (with the typical immersion time being 90 min) at standard temperature and pressure inside a glass scintillation vial that was shielded from light with aluminum foil (see Supplementary Fig. 2). After immersion, the samples were rinsed with deionized water and dried with nitrogen flow.

### Characterization

Confocal PL spectroscopy, Raman spectroscopy, and PL mapping in ambient conditions were performed with a Horiba XploRA Plus instrument using a 532 nm laser focused with an objective (×100, NA 0.9). AFM imaging was performed with an Asylum Cypher S instrument in tapping mode using NCHR-W Pointprobe tips. The spring constant of the cantilever was estimated by relating the nominal value of the spring constant and resonance frequency from the manufacturer and the measured resonance frequency using the following relation[14]:

$$k_{\text{estimated}} = k_{\text{nominal}} \left( f_{\text{measured}} / f_{\text{nominal}} \right)^3 \qquad (1)$$

The cantilever was calibrated using thermal tuning to determine the inverse optical lever sensitivity factor (InvOLS) to allow conversion of the deflection voltage into cantilever deflection distance. With a

contact-mode setpoint of 0.05–0.2 V, AFM scratching was performed with an estimated normal force setpoint of 50–200 nN. XPS measurements were performed in high vacuum (-1 × 10⁻⁸ mbar base pressure) using a Thermo Scientific ESCALAB 250 Xi instrument in charge compensation mode with a nominal spot size diameter of 400 μm. The binding energy was calibrated to the adventitious C 1 s level at 248.8 eV.

### Low-temperature optical spectroscopy

Low-temperature optical spectroscopy measurements (Figs. 2 and 3 and Supplementary Figs. 6, 9–10, 12) were conducted in a closed-cycle cryostat (Attocube, AttoDRY2100) at a temperature of 1.7 K with a superconducting magnet (see Supplementary Fig. 32). For confocal PL measurements, a tunable CW laser (M Squared, SolsTis EMM) at $\lambda = 635$ nm was used, while a broadband white light source (Thorlabs, SLS201) was used for reflectivity measurements. A 100× magnification objective with a 0.82 NA (Attocube, LT-APO/VIS/0.82) was used to focus the laser beam or white light and collect the PL or reflection signal. We estimate the diffraction-limited beam diameter of the PL measurement to be $D = 1.22\lambda/N.A. \approx 1$ μm. The excitation powers used were 60 μW for PL spectroscopy and $g^{(2)}(\tau)$ measurements in Fig. 2, 10 μW for the PL map and spectra in Fig. 3 before functionalization, 45 μW for PL map and spectra in Fig. 3 after functionalization, 10 μW for PL spectrum in Fig. 3g, and 10 μW for the $g^{(2)}(\tau)$ measurement in Fig. 3h. Typically, the emitters both before and after functionalization achieve intensity saturation for excitation powers below 10 μW.

The collected signal was sent to a spectrometer of 750 mm focal length (Andor Shamrock, SR-750-D1-R-SIL) equipped with a thermoelectrically-cooled CCD camera (Oxford Instruments, DU420A-BEX2-DD). The PL maps were obtained by scanning the cryogenic non-magnetic linear nanopositioners (Attocube, ANPx101/RES/LT) in the $x$ and $y$ directions. PLE measurements were performed using the TeraScan function of an ultranarrow linewidth CW Ti:Sapphire laser (M Squared, SolsTis). A Hanbury Brown–Twiss setup was used to measure the second-order correlation function. The emission signal was filtered by a band-pass filter (10 nm FWHM bandwidth) to block the emission other than the desired SPE, where an adjustable wavelength range is achieved by mounting the filter at a tilt angle. The filtered signal was then coupled to a fiber connected with a 1 × 2 fiber splitter to split the emission and direct the signal to two avalanche photodiodes (APD; PicoQuant, τ-SPAD-100).

The raw coincidence data ($g_{raw}^{(2)}(\tau)$) were corrected for the background to obtain the correlation function according to the relation[56]

$$g^{(2)}(\tau) = \frac{g_{raw}^{(2)}(\tau) - (1 - \rho^2)}{\rho^2} \qquad (2)$$

where $\rho = S/(S + B)$ is related to the signal-to-background ratio. Background correction is a commonly performed procedure in the analysis of the second-order autocorrelation function. In the case of solid-state quantum emitters, background correction has been applied to diverse systems, including monolayer WSe₂[5,57,58], hBN[59–61], diamond[56,62–64], and SiC[65]. Our method of background correction is consistent with this prior literature. The correction of the coincidence count was made by accounting for only the background count ($B$) that arises from contribution of the dark count of the APD detector and environmental signal that is sample-independent. This background count was defined as the count measured by the APD while blocking the laser excitation from reaching the sample. No other correction of the $g^{(2)}(\tau)$ data (e.g., background subtraction from uncorrelated portions of the emission spectrum, assumptions about instrumental jittering, deconvolution from the instrument response function, or symmetrization) was performed.

Low-temperature PL lifetime microscopy measurements (Supplementary Fig. 5) were performed using an optical setup consisting of a long-pass dichroic mirror (650 nm), a mirror mounted on a scanning S-335 Piezo platform (Physik Instrumente), a scan lens (Thorlabs LSM03-VIS), a 100 mm tube lens (Thorlabs TTL100-A), and an optical cryostat (Montana Instruments Cryostation S100) with a built-in objective (Zeiss Epiplan-Neofluar 100x/0.90 NA). The sample was excited with a picosecond pulsed laser (LDH-P-C-640B, adjusted to 10 or 20 MHz, Picoquant PDL 800-D) and the collected PL signal was measured with an avalanche photodiode detector (Micro Photon Devices PDM) in a confocal arrangement. The PL signal was separated from the excitation beam with a 700 nm long-pass filter, while the SPE signal was filtered using a combination of short- and long-pass filters. The IRF of the system is -95 ps.

### First-principles calculations

1L WSe₂ was modeled using a $4 \times 4 \times 1$ supercell. The NPh oligomers were modeled using different numbers and geometries of NPh rings as shown in Supplementary Figs. 23 and 24. These isolated molecules were then noncovalently interfaced to 1L WSe₂. All calculations were performed using the Vienna ab initio simulation package (VASP) based on spin-polarized density functional theory (DFT) with a plane-wave basis set and projector-augmented wave (PAW)[66] technique. For geometry optimization, the generalized gradient approximation (GGA) refined by Perdew, Burke, and Ernzerhof (PBE)[67] was utilized with Grimme's DFT-D3BJ correction. The energy and force convergence parameters were $1 \times 10^{-6}$ eV/cell and $1 \times 10^{-2}$ eV/Å, respectively. The energy cutoff was 520 eV, and the Brillouin zone was sampled using the Γ-centered Monkhorst–Pack k-grid scheme with a $4 \times 4 \times 1$ k-mesh. A 20 Å vacuum above the surface along the $c$-axis was added to avoid inter-image interactions. In the most stable configuration, the NPh oligomers were flat to the surface. In contrast, the vertical orientation of NPh relative to the WSe₂ surface[68] had a Se–C bond length of 2.54 Å, which was energetically less stable by 0.26 eV compared to the flat configuration. The isolated molecules were simulated employing a cubic box of $50 \times 50 \times 50$ Å and at the Γ point ($1 \times 1 \times 1$). The binding energy of the modeled NPh oligomer on the surface was defined as:

$$E_{binding} = E_{total} - (E_{WSe_2} + E_{NPh}) \qquad (3)$$

where $E_{total}$, $E_{WSe_2}$, and $E_{NPh}$ denote the total energy of the system, the energy of 1L WSe₂, and the isolated NPh oligomer energy, respectively. The binding energy for the different NPh oligomer configurations is provided in Supplementary Table 2, where negative binding represents an exothermic reaction. Biaxial and isotropic tensile strains of 1% and 3% were applied to the pristine, Se mono-vacancy (Se$_{vac}$), W mono-vacancy (W$_{vac}$), and NPh functionalized (with and without vacancy) surfaces along the $a$ and $b$ axes (comprehensive results are provided in Supplementary Figs. 27–29). All of the band structures along the high symmetry points in the Brillouin zone and the energy level alignments were obtained using the range-separated hybrid functional HSE06[69] coupled with spin–orbit coupling (HSE06/SOC) on the PBE-D3BJ geometry. Band structures and other post-processing were carried out using VASPKIT[70].

## Data availability

Relevant data supporting the key findings of this study are available within the article and the Supplementary Information file. All raw data generated during this study are available from the corresponding authors upon request.

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

## Acknowledgements

This research was primarily supported by the Center for Molecular Quantum Transduction, an Energy Frontier Research Center funded by the U.S. Department of Energy, Office of Science, Office of Basic Energy Sciences, under Award No. DE-SC0021314. This work was performed, in part, at the National Science Foundation Materials Research Science and Engineering Center at Northwestern University under Award No. DMR-1720319. Work performed at the Center for Nanoscale Materials, a U.S. Department of Energy Office of Science User Facility, was supported by the U.S. DOE, Office of Basic Energy Sciences, under Contract No. DE-AC02-06CH11357. This work also made use of the Keck-II and EPIC facilities of the Northwestern University NUANCE Center, which has received support from the SHyNE Resource (NSF ECCS-2025633), the IIN, and the Northwestern MRSEC program (NSF DMR-1720139). K.W. and T.T. acknowledge support from JSPS KAKENHI (Grant Numbers 19H05790, 20H00354 and 21H05233). A.D. acknowledges a National Science Foundation Graduate Research Fellowship. D.L. acknowledges support from the Swiss National Science Foundation for an Early Post-Doc Mobility Fellowship (P2EZP2_181614). The authors also acknowledge Roel Tempelaar, Teodor K. Stanev, and Pufan Liu for useful discussions.

## Author contributions

M.I.B.U., A.D., D.L., W.W., K.W., and T.T. contributed materials and sample preparation. M.I.B.U. and A.D. performed material characterization at room temperature. H.Z., S.C.G., R.A., J-S.C., and W.W. performed low-temperature optical spectroscopy. T.S. performed first-principles calculations. M.I.B.U. and H.Z. analyzed the data. M.I.B.U. prepared the manuscript with input from all authors. M.C.H., N.P.S., G.C.S., E.A.W., X.M., and T.J.M. supervised the project.

## Competing interests

The authors declare no competing interests.
