## [Peer Review File · Nature Communications]

Chemomechanical Modification of Quantum Emission in Monolayer WSe₂Editorial Note: This manuscript has been previously reviewed at another journal that is not operating a transparent peer review scheme. This document only contains reviewer comments and rebuttal letters for versions considered at *Nature Communications*.

Point-by-point response to reviewer comments

Reviewer #1

Comment:

After carefully reading the replies to all referee comments and the updated manuscript, my opinion is unfortunately unchanged: I do not find the concept in the manuscript likely to produce impact in the community. The simple reason is that there is no evidence the quantum emitters resulting AFTER the treatment are any more useful than before the treatment. There may be fewer of them, which can be helpful, but this is an objective the community already knows how to achieve without chemical modification. But, the properties of the emitters are not any more attractive. In fact, the properties of these emitters are not at the state of the art level: the linewidths are worse (many reports of 30-100 micro-eV linewidths in literature compared to the several hundred micro-eV linewidths here), the count rates are quite modest (few kHz compared to 100's of kHz in literature), and the g₂ values are quite standard (and background subtraction is never justified in my opinion – its data manipulation based on assumptions).

Response:

We thank the reviewer for the careful consideration of our work. In addition to demonstrating the benefits of the 4-NBD treatment for producing better isolated emitters in 2D materials, our work represents a conceptual advance that introduces a new parameter space for modifying the properties of single-photon emitters in 2D materials, particularly highlighting the prospects of using interfacial engineering via chemical functionalization.

To elaborate further on our previous response, the chemical functionalization approach is compatible with other schemes and strategies that have been previously described in the literature to achieve state-of-the art emitters in terms of the brightness and linewidth. This compatibility is facilitated by the fact that chemical functionalization requires access to only one side of the WSe₂ monolayer surface. Specific to the 4-NBD treatment, since the linewidth and brightness of emitters before and after functionalization are comparable, it is highly likely that improvements for the emitter properties before functionalization would be maintained after 4-NBD treatment.

For example, reduction in emitter linewidth can be achieved by using high-quality substrates and straining structures with lower defect and interface states than that of SiO₂ used in our work. Previous work that has demonstrated narrow linewidths reaching < 100 μeV in WSe₂ include the use of substrates or capping of Al₂O₃ (which can then be proximitized with metallic structures for plasmonic coupling effect, e.g., Nat Nanotech 2018, 13, 1137; ACS Photon 2018, 5, 1919), InGaP (e.g., Optica 2017, 4, 669), or hBN (e.g., Nat Nanotech 2019, 14, 426; Nat Commun 2021, 12, 3585). Meanwhile, enhancement of the emitter brightness can be achieved by integrating the chemomechanically-modified emitters with plasmonic structures (e.g., Nat Nanotech 2018, 13, 1137, ACS Photon. 2018, 5, 3466) or by relying on enhancement effects with photonic structures (e.g., Nano Lett. 2021, 21, 4715; Nat Commun 2021, 12, 6063; APL 2018, 112, 191105).

Alternatively, we can envision that further permutations of the chemomechanical strategy may also achieve improvements in emitter properties by exploring other combinations of molecules and 2D materials. For example, selecting molecules with precisely aligned energy levels could improve the selectivity of the

emitter wavelength toward a longer wavelength range or to a very narrow spectral range, which could be valuable in reducing the inhomogeneous broadening of these emitters. Likewise, improvement in emitter brightness may be achieved by selecting molecules that form type-I heterojunction with 2D materials (e.g., Adv. Mater. 2018, 30, 1803986)

Regarding concerns about background correction, we note that this procedure is commonplace for the analysis of the second-order autocorrelation function. In the case of solid-state quantum emitters, background subtraction was discussed and employed by various investigators working on systems including monolayer WSe₂ (e.g., Nature Commun. 2019, 10, 4435; Optica 2016, 3, 882; Optica 2015, 2, 347; Nano Lett. 2015, 15, 7567), hBN (e.g., Nanotechnology 2022, 33, 015001; Nano Lett. 2017, 17, 2634; Phys. Rev. B 2018, 98, 081414(R)), diamond (Nature Commun. 2019, 10, 2392; Phys. Stat. Sol. A 2013, 210, 2060; Eur. Phys. J. D 2002, 18, 191; Opt Lett 2000, 25, 1294), and SiC (Nature Mater. 2015, 14, 160).

Our method of background correction is consistent with this prior literature. The correction of the coincidence count was made by accounting for only the background count that arises from contributions of the dark count of the APD detector and environmental signals that are sample-independent. This background count was defined as the count measured by the APD while preventing the laser excitation from reaching the sample. No other correction of the $g^{(2)}(\tau)$ data (e.g., background subtraction from uncorrelated portions of the emission spectrum, assumptions on instrumental jittering, deconvolution from the instrument response function, or symmetrization) was performed.

In the revised manuscript, we have:

- Included a discussion in the conclusion section on how chemical functionalization of 2D materials can be incorporated with other existing strategies to achieve state-of-the-art emitter performance in terms of linewidth and brightness.
- Included the discussion of the background correction for $g^{(2)}(\tau)$ in the methods section of the main text.
- Combined the raw $g_{\text{raw}}^{(2)}(\tau)$ and the background-corrected $g^{(2)}(\tau)$ data in the same graphs with their values shown on the left and right y-axes, respectively.

Reviewer #2

General comment:

The authors have adequately responded to all of my comments, and I believe the paper is strengthened by their revisions and additional discussion. I believe the community working on 2D quantum emitters will find this work interesting and potentially useful for engineering emitters in WSe₂. I believe it will also lead to additional follow-up experiments and modeling to understand how to further improve the control of the emitters' properties. I am happy to recommend publication.

Response:

We thank the reviewer for the interest in our work and for recommending the publication of our manuscript.

Reviewer #3

General comment:

The authors provided an extensive response to all the referees' remarks. Although some of these points could be addressed more in detail in the revised version of the manuscript, the correction is still satisfactory. Given the novelty of the work, it is not surprising that the approach and results bring about several questions. I believe that the paper in its present form fulfills the standards of Nature Communications and will be of interest to the journal's broad readership.

Response:

We thank the reviewer for appreciating the novelty of our work and for recommending the publication of our manuscript.

Reviewer #4

General comment:

The authors have improved the manuscript in response to the revisions. I now understand the proposed mechanism of “SPE creation” after 4-NBD treatment better. Namely, the authors claim that most bright emission channels are quenched by the treatment, making way for emission from channels that were previously being outcompeted by the initially bright channels. While this mechanism seems plausible, I don’t think the data provides strong evidence that this is the case.

The concept of using a chemical treatment to selectively quench emission in certain energy ranges is an exciting idea. I think the experiments and model provide decent evidence that this could be the case. I think much stronger evidence would be to show that different chemical treatments quench different ranges of energies, depending on their energy levels relative to WSe₂, but I recognize that this is likely outside of the current scope of this work.

Overall, I think the idea of using chemical treatment to modify the properties of SPEs in TMDs is novel and potentially useful. The observation of cleaner spectra after treatment seems robust. The demonstration of non-covalent bonding also seems robust. I am not thoroughly convinced that the proposed mechanism is correct, but I think the proposed mechanism is generally consistent with the data.

Response:

We thank the reviewer for the positive assessment of our manuscript and for appreciating our chemical functionalization approach.

Comment 1:

I have a few detailed comments below, following up on my previous round of comments.

One thing that I find slightly misleading in the manuscript is that the authors used significantly more power to characterize the SPEs after 4-NBD treatment than before, which gives the impression that the emitters are brighter after 4-NBD treatment than before, but I’m not sure this would be the case with the same input power. The authors claim that the SPEs saturate at a few uWs based on Fig. 2f, but it looks to me like the intensity continues to change significantly (albeit sublinearly) up to 40 uW. Since emitter brightness is an important factor, I think it is important to more carefully explain the comparison of emitter intensity before and after 4-NBD treatment.

Even considering the larger power used for Fig. 3 after treatment, I’m still confused why the neutral exciton is so big after the treatment relative to before. In Fig. 2b and c, the neutral exciton is ~2-3x bigger before treatment with similar power used before and after treatment. However, in Fig. 3f, the neutral exciton is barely visible or not visible at all before treatment while the neutral exciton is pretty big after treatment. Why does the neutral exciton not show up significantly before treatment on the pillars, but it is quite strong after treatment on pillars? (I understand that the power was higher after treatment, but if you divided the red spectra by 45/10, it would still have an observable neutral exciton peak.) Is the difference between Fig. 2 and Fig. 3 related to the presence of the pillar?

Response:

We thank the reviewer for this comment. The original purpose of the low temperature PL measurements (including Fig. 3 for the sample on nanopillars) was to demonstrate the primary effect of 4-NBD treatment in partially quenching defect-related emission lines, resulting in the simplified emission spectrum. With this purpose in mind, measurements with different excitation power can be compared since the primary properties of interest are the lineshape changes of the emission spectrum (e.g., with the normalized PL intensity as was the case in the original version of the submitted manuscript). Relatedly, the higher excitation power used after 4-NBD treatment is also beneficial for confirming that the emitter lines are indeed quenched.

On the other hand, we agree with the reviewer that an exact comparison of the emitter brightness is less ideal with the different excitation conditions, although we have argued that at a power level of 10 μW or above the emitter intensity tends to be close to saturation. Therefore, as an additional experimental observation, we present here another low temperature experiment on a different sample where the measurements before and after 4-NBD treatment were performed by using the same excitation power (46 μW). This experiment facilitates a more direct comparison of the typical intensity of the defect-related emitters before and after 4-NBD treatment (Fig. D1).

Fig. D1 | Additional low temperature PL measurements on WSe₂ monolayer flakes transferred on a Si/SiO₂ substrate. For both before and after 4-NBD treatment, PL measurements were performed with 635 nm CW laser excitation at 46 μ W power at T = 2.8 K (Attocube AttoDRY2100). **a**, Optical micrograph of the sample. **b,c** AFM image of the sample (**b**) before and (**c**) after 4-NBD treatment, showing wrinkle formation at the corner of the flake. **d**, PL map of the sample before functionalization. The images show the maximum count for each pixel from the same PL map for emission wavelength ranged longer than that of the neutral exciton, X₀. (left) Emission wavelength > 730 nm. (right) Emission wavelength range > 770 nm. The bright spots from the two maps experienced enhanced defect-related emission from randomly-occurring localized strain and are labelled as “Wrinkle1” and “Wrinkle2”. **e**, PL map after 4-NBD treatment, showing the maximum PL count for emission wavelength range > 730 nm. **f**, PL spectra before and after 4-NBD treatment from the flat region of the sample (left) and locations on the map indicated as Wrinkle1 (middle) and Wrinkle2 (right). At Wrinkle1 location, the spectrally dense defect emissions are bundled together into a single high intensity broad feature between 720 nm and 760 nm. The insets show the spectra in logarithmic y-axis for better visualization. Simplified emission spectra after 4-NBD treatment are observed, reducing the number of defect-related emitter lines.

In this experiment, we reproduced the main result of the manuscript, namely the simplification of the emission spectrum following the 4-NBD treatment. At sample locations that experience enhanced defect-related emission due to localized strain (Wrinkle1 and Wrinkle2 in Supplementary Fig. D1d-f), the 4-NBD treatment results in partial quenching, resulting in emission lines that are better isolated energetically. The remaining emitter shows a peak intensity of ~100-500 kHz, which is comparable to the emitters before functionalization (most visible in Wrinkle2 location).

In the meantime, the absolute intensity of the neutral exciton peak (X₀) may show some variation from sample to sample. Likewise, the amount of relative quenching of X₀ intensity before and after 4-NBD treatment may also vary from the flat region of the sample to the region of the sample with localized strain. For example, the quenching of the X₀ peak at the location Wrinkle2 appears more dramatic than that of the flat region, despite the defect-related feature at Wrinkle 2 remaining of comparable intensity. As the reviewer suggested, it is possible that the effect may also vary at different nanopillar-strained locations of the sample in such a way that the quenching level of X₀ may appear attenuated. The variation of measured intensity of X₀ at such strained locations is also complicated by the fact that the emission involves multiple defect-related recombination decay channels, and thus future work will explore time-resolved measurements to deconvolve these dynamics.

In the revised manuscript, we have included Fig. D1 and the associated discussion as Supplementary Fig. 13 in the SI.

Comment 2:

Regarding the following statement: “The 4-NBD treatment effectively quenches these defect-related emission features, but also introduces a sharp, spectrally-isolated PL peak near 788 nm (Fig. 2c, red).” ...I think this statement is misleading and suggests that the 4-NBD treatment creates an emitter site rather than changing conditions to reveal a site that already existed, which is what I understand as the proposed mechanism. I think the authors should revise this language to avoid confusion.

Response:

We thank the reviewer for the suggestion. We have revised the relevant passage in the manuscript to the following:

“The 4-NBD treatment effectively quenches these defect-related emission features (Fig. 2c, red), resulting in significantly fewer emission features that are spectrally-isolated (e.g., the sharp peak near 788 nm). As we shall discuss later, it is likely that the emission features remaining after 4-NBD treatment also originate from pre-existing defects within the WSe₂ monolayer.”

REVIEWERS' COMMENTS

Reviewer #4 (Remarks to the Author):

The authors have satisfactorily addressed my concerns and I think this paper is suitable for publication in this journal. I think the chemical approach to modifying 2D material SPE behavior is novel and will be of interest to researchers in this relatively broad area.